# Effect of Cysteine with Essential Oils on Quality Attributes and Functional Properties of ‘Blanca de Tudela’ Fresh-Cut Artichoke

**DOI:** 10.3390/foods12244414

**Published:** 2023-12-08

**Authors:** María J. Giménez, Marina Giménez-Berenguer, Fabián Guillén, Vicente Serna-Escolano, María Gutiérrez-Pozo, Pedro J. Zapata

**Affiliations:** Department of Food Technology, Escuela Politécnica Superior de Orihuela, University Miguel Hernández, Ctra. Beniel km 3.2, 03312 Alicante, Spain; maria.gimenezt@umh.es (M.J.G.); marina.gimenezb@umh.es (M.G.-B.); fabian.guillen@umh.es (F.G.); vserna@umh.es (V.S.-E.); pedrojzapata@umh.es (P.J.Z.)

**Keywords:** artichoke, browning, sensory analysis, total phenolics, total antioxidant activity

## Abstract

The commercialisation of fresh-cut artichokes with optimal quality and appearance and a maximum shelf-life is a great challenge for the artichoke market. The use of different anti-browning agents has been previously studied; however, their effect is still limited. Therefore, the objective of this study is the evaluation of the effect of L-cysteine and, in combination with a mixture of essential oils components (eugenol, thymol and carvacrol) on browning, quality and bioactive compounds of fresh-cut artichokes stored for 9 days at 2 °C. Four different treatments were applied to ‘Blanca de Tudela’ fresh-cut artichokes: cysteine and cysteine with 75, 150 and 300 µL of the essential oils components (EOs) mixture. After 2, 4 and 9 days of storage, physicochemical parameters (weight loss, colour, respiration rate) and functional (total phenolic content, antioxidant activity) were studied. A descriptive sensorial analysis was also carried out to evaluate sensory attributes. Results showed that the application of cysteine and 150 µL of EOs displayed the lowest browning and highest antioxidant properties, as well as the best quality and sensory parameters. The use of this post-harvest treatment on fresh-cut artichokes would result in a natural and eco-friendly solution to improve artichoke quality and shelf-life.

## 1. Introduction

Artichoke (*C. scolymus* L.) is widely cultivated around the world, with Italy, Egypt, and Spain being the three main producers. Spain is in the second position in Europe, generating around 200 million tons of artichokes per year [1]. It is an essential component of the Mediterranean diet due to its high content of bioactive compounds, fibre and minerals [2]. It is considered a highly perishable commodity that suffers several physiological changes during its storage, resulting in a great inconvenience for its marketability. Nowadays, fresh-cut artichokes with pleasant appearances and organoleptic and nutritional characteristics are requested by consumers. This is a great challenge for the artichoke market, where optimal quality and a longer shelf-life are required [3].

The main causes of quality losses in fresh-cut artichokes are enzymatic browning, weight loss, dehydration and physical injuries, such as bruising or compression. Enzymatic browning is considered the main cause of quality loss in fresh-cut artichokes [3,4]. It takes place due to the high polyphenol content of artichokes, polyphenols that serve as substrate for the polyphenol oxidase (PPO). Once artichokes are subjected to wounding or cutting, a cellular disruption takes place, allowing PPOs to make contact with their substrates. These polyphenols are oxidated to quinones, resulting in dark pigment causing tissue browning on fresh-cut artichokes [5,6,7]. The use of low phenolic content cultivars for the minimally processed market (fresh-cut artichokes) has been previously studied [8,9]. The effect of harvest time and storage conditions (time and temperature) on the total phenolic content of artichokes and, therefore, their effect on the enzymatic browning of fresh-cut artichokes [10,11,12,13,14] has been previously studied. It has been observed that the use of Modified Atmosphere Packaging (MAP) combined with low storage temperatures [15,16,17] and in combination with anti-browning agents, edible coatings or innovative packaging [17,18,19,20,21,22] significantly reduces enzymatic browning and increases the shelf-life of fresh-cut artichokes.

The application of anti-browning treatments, such as ascorbic acid, oxalic acid, citric acid or L-cysteine, as excellent antioxidants have been widely studied [10,23,24,25,26]. L-cysteine has been selected as the most effective anti-browning treatment for fresh-cut artichoke due to its inhibition effect on the polyphenol oxidase (PPO). When oxygen is available, PPO catalyses the oxidation of phenols, leading to the appearance of browning pigments [27,28]. Several authors have observed a reduction in browning and an extension of the shelf-life of fresh-cut artichokes when they were treated with L-cysteine. However, an increase in the yellow colour of fresh-cut artichokes was also observed as a result of the application of amino acids [23,24,29].

Therefore, research about new alternatives to L-cysteine to reduce browning and extend shelf-life in fresh-cut artichoke is required. Previous studies have elucidated the potential antioxidant effect that essential oils’ major components (EOs) have when they are applied to fruits and vegetables at different stages of the food chain [30,31]. EOs contain secondary metabolites produced in different parts of the plant, known to confer disease resistance, antioxidant and antimicrobial properties [32,33,34]. Their use as part of edible coating has been extensively studied to reduce weight loss, respiration rate and fungal decay in citrus fruit [35,36,37,38] and table grapes [39,40], among others. The synergic effect that major components of essential oils (eugenol, thymol, and carvacrol) have on the fungal spoilage of fruit has been previously demonstrated [38,41]. Its use for the minimally processed market as an edible coating is still reduced; only a few studies have been carried out on the use of an antimicrobial agent in fresh-cut apples [42] and fresh-cut melon [43]. Rizzo et al. [44] have studied the effect of locust bean gum edible coating with added *Foeniculum vulgare* EO and its use as an active packaging-releasing system in fresh-cut artichokes as an anti-browning treatment. However, most of the studies are focused on the effect of these EOs on the sensorial and colour parameters of treated fresh-cut artichokes, while parameters such as weight loss, respiration rate and firmness are also required to achieve the highest quality in fresh-cut artichokes.

Hence, this study aimed to evaluate the effect of L-cysteine and, in combination with a mixture of essential oils components (eugenol, thymol and carvacrol) on browning, quality and bioactive properties of fresh-cut artichokes stored for 9 days at 2 °C.

## 2. Materials and Methods

### 2.1. Plant Material and Experimental Design

Artichokes (*C. scolymus* L. cv. ‘Blanca de Tudela’) were harvested from a commercial plot located in Orihuela (Alicante, Spain). Heads were harvested when fully developed according to standard practices and immediately transported to the laboratory. Selection and processing were performed at 10 °C under hygienic conditions. Artichokes with defects were discarded, and heads were trimmed using a sharp stainless-steel knife to remove external greener and tougher bracts (inedible fraction) and the upper portion. After that, each artichoke heart was cut into 8 parts (1/8), mixed, and dipped in ice water containing 100 ppm of sodium hypochlorite (Sigma Aldrich, Madrid, Spain) as a disinfectant solution. The essential oils components (EOs) for this study were eugenol, thymol and carvacrol. All heart slices were drained and 72 different 1/8 parts were randomly selected for each different treatment and replicates and dipped in distilled water (Control), L-cysteine at 0.028 M at a pH of 2.2 (Cys), cysteine + 75 µL/L EOs (Cys + 75 EOs: 25 µL eugenol + 25 µL thymol + 25 µL carvacrol), cysteine + 150 µL/L EOs (Cys + 150 EOs: 50 µL eugenol + 50 µL thymol + 50 µL carvacrol), cysteine + 300 µL/L EOs (Cys + 300 EOs: 100 µL eugenol + 100 µL thymol + 100 µL carvacrol). The reagents used were L-cysteine hydrochloride monohydrate, carvacrol (>98%), eugenol (>98%) and thymol (98.5%) (Sigma Aldrich, Madrid, Spain) at the specific concentrations diluted in distilled water. For each treatment, artichoke slices were divided into lots and dipped for 1 min in the corresponding solution and then dried at room temperature and stored in polyethylene (PE) opened trays at 2 °C and 85% RH for 9 days. Samples of eight 1/8 heart parts from each treatment in triplicate were evaluated for quality and bioactive attributes.

### 2.2. Artichoke Quality Parameters

Quality parameters were evaluated after 2, 4 and 9 days at 2 °C. These days represented an early, mid- and late stage of the storage of fresh-cut artichokes. Individual replicates were weighted daily, and Weight Loss (WL) was expressed as a percentage (%). The respiration rate (RR) was measured for the three replicates of each treatment. Eight 1/8 heart slices of artichokes were placed in a 1 L glass jar hermetically sealed for 30 min at room temperature, and 1 mL of the headspace was analysed and quantified in a gas chromatograph (TMGC-2010, Shimadzu Corporation, Kyoto, Japan) equipped with a thermal conductivity detector (TCD). Results were expressed as the mean ± standard error (SE) in mg of CO_2_ kg^−1^ h^−1^. Browning of artichokes was evaluated individually based on colour parameters (L*, a* and b*), using a Minolta colourimeter (CRC200; Minolta, Osaka, Japan) at three points of the external surface for each artichoke part using the CIE Lab System. Results were expressed as the mean ± SE.

### 2.3. Total Antioxidant Activity

Total Antioxidant Activity (TAA) was quantified according to Valero et al. [45], where both hydrophilic (H-TAA) and lipophilic (L-TAA) compounds were determined. Eight parts (1/8) of artichoke heads for each treatment were cut into small pieces, and three replicates of five grams each were weighed for the following extraction. Artichoke pieces were homogenised in 15 mL of 50 mM Na-PO4 buffer (pH 7.8, Sigma-Aldrich, Madrid, Spain) and 10 mL of ethyl acetate and then centrifuged at 10,000 × g for 15 min at 4 °C. The L-TAA was determined from the upper fraction, while the lower fraction was used for H-TAA quantification. The quantification was based on the enzymatic system composed of ABTS, peroxidase enzyme (HRP) and its oxidant substrate, hydrogen peroxide (H_2_O_2_), in which ABTS+ radicals were generated and monitored at 730 nm. The reaction was carried out in duplicate following the Re et al. [46] protocol with some modifications, as presented in Giménez et al. [47]. The results were expressed as the mean ± SE of mg of Trolox equivalent per 100 g of fresh weight (FW).

### 2.4. Total Phenolic Content

Total Phenolic Content (TPC) was measured following Swaim and Hillis [48] protocol with slight modifications, as previously mentioned by Martínez-Esplá et al. [49]. A water: methanol (2:8) solution containing 2 mM NaF (Sigma-Aldrich, Madrid, Spain) was used, and extracts were quantified using the Folin–Ciocalteu reagent. Results (mean ± SE) were expressed as mg of gallic acid equivalent per 100 g FW.

### 2.5. Descriptive Sensory Analysis

The descriptive sensory analysis of fresh-cut artichokes was performed by a panel of 10 judges (5 males and 5 females) trained in sensory testing from the Department of Agri-Food Technology of Miguel Hernández University (Orihuela, Alicante, Spain). Two preliminary sessions were carried out to discuss the attributes considered by the panellist that could provide a more suitable description of the fresh-cut artichokes. The sensory attributes (visual appearance, odour, taste, texture) evaluated are presented in Table 1, according to García-Martínez et al. [16]. The panellists used a hedonic scale from 0 to 5, previously designed by Giménez et al. [19], where 1 to 2 is very poor, 3 to 4 is fair, and 5 is excellent. A score of 3 or less for any of the organoleptic characteristics evaluated indicated the end of the shelf-life.

### 2.6. Statistical Analysis

Data sets of the different evaluated parameters (weight loss, respiration rate, colour, TPC and TAA) were subjected to an analysis of variance (ANOVA) to determine significant differences between treatments (*p*-value < 0.05). When significant differences were detected, a post hoc analysis was carried out using HSD Tukey’s test for each pair of treatments. All statistical analyses were performed using SPSS software v. 20.0 for Windows.

## 3. Results and Discussion

### 3.1. Effect of the Application of Cysteine and EOs on Artichoke Browning

Fresh-cut artichokes are susceptible to suffering browning during cold storage, generating an undesirable appearance, causing rejection by consumers and limiting their shelf-life. Browning is caused by the oxidation of phenolic compounds catalysed by different enzymes, such as polyphenol oxidases (PPO), with the subsequent formation of dark compounds [4,8,50]. In the present work, browning was studied in non-treated and treated artichoke slices based on appearance (Figure 1) and colour parameters (Figure 2a,b). The appearance of artichokes considerably improved when they were treated with a combination of cysteine and EOs, observing the best appearance in those artichokes that were treated with cysteine and 150 µL of EOs (Cys + EOs 150). In addition, both L* (lightness) and b* (yellowness) colour parameters were studied in artichoke slices after 2, 4 and 9 days of storage at 2 °C. After four days of storage, the L* value of non-treated artichokes was 72.67 ± 0.90, while Cys, Cys+ 75 EOs, Cys + 150 EOs, and Cys + 300 EOs treated artichokes were 81.05 ± 0.45, 81.48 ± 0.42, 82.35± 0.43 and 81.39 ± 0.49, respectively (Figure 2a). Significant differences (*p*-value < 0.05) were detected between treated and non-treated artichokes after 2 and 4 days of storage. After 9 days, significant differences were only detected in those artichokes treated with cysteine and 150 and 300 µL of EOs. Contrary results were observed by Rizzo et al. [51], who found that the use of *Foniculum vulgare* EO did not influence the colour appearance of fresh-cut artichokes. It has been previously reported that artichokes’ quality characteristics depend on cultivar, genotype, harvest, and storage time [8,9,11,50]. Furthermore, in a previous study, ‘Blanca de Tudela’ fresh-cut artichokes were treated with different concentrations of cysteine (10–50 mol/m^3^) and packed in polypropylene films, achieving L* values of 60.17 ± 0.22 after 4 days at 5 °C [29], while in the present study, the combination with EOs resulted in a 28% higher lightness. A decrease in L* with storage time was observed in all treatments and has been previously reported [44,51]. It is associated with the PPO enzymatic reaction that occurs during the browning process [52,53]. The application of Cys + 150 EOs on artichokes slices resulted in a 5% reduction in lightness with time, while for non-treated fresh-cut artichokes, a reduction of 16.31% was observed.

On the other hand, significant differences (*p*-value < 0.05) were also detected in the yellowness (b*) between treated and non-treated artichokes (Figure 2b). The increase in lightness in those artichokes treated with cysteine has been previously reported by several authors [23,24,29]. Significantly higher values of b* were observed in those artichokes treated with cysteine and 150 and 300 µL of EOs after 2 and 4 days of storage, while after 9 days of storage, significant differences were detected between treated and non-treated artichokes, but no differences were observed between treatments, observing a b* of 31.38 ± 0.48 in the non-treated artichokes and values of 45.84 ± 0.97, 45.58 ± 0.97, 45.28 ± 0.51 and 45.42 ± 0.94 in Cys, Cys+ 75 EOs, Cys + 150 EOs and Cys + 300 EOs treated artichokes. In a previous study, Amodio et al. [23] treated fresh-cut Catanese cv artichokes with 0.25% and 0.5% of cysteine and stored them for 5 days at 5 °C, observing L* values of 33.4–38.4 and b* values of 36.3–38. While in the present study, the application of cysteine and EOs resulted in higher lightness and yellowness.

Therefore, the use of cysteine in combination with EOs significantly increased lightness and yellowness and reduced the browning of fresh-cut artichokes; consequently, a considerable improvement in appearance was achieved.

### 3.2. Effect of the Application of Cysteine and EOs on Quality Parameters of Fresh-Cut Artichokes

Weight loss (WL) during the transpiration process of artichokes is an important cause of deterioration affected by temperature and reduces product quality when artichokes are sold as processed products [54]. Weight losses increased during storage in fresh-cut artichokes, independently of treatment (Figure 3a). However, a lower increase with time was observed in those fresh-cut artichokes that were treated with the combination of cysteine and EOs. The use of cysteine and EOs at 75 and 150 µL significantly (*p*-value < 0.05) reduced the weight loss of fresh-cut artichokes compared to the rest of the treatments during the storage period. After 9 days of storage at 2 °C, Cys + 75 EOs and Cys + 150 EOs treated artichokes presented an 18.73 (19.41 ± 0.47%) and 20% (19.72 ± 0.75%) reduction of WL compared to the non-treated artichokes (24.26 ± 1.61%), respectively. In previous studies, the use of *Foeniculum vulgare* EOs on its own or in combination with cysteine observed a similar WL when artichokes were hermetically packed in PET trays [44,51]. In the present study, a decrease in the WL has been observed, probably due to the absence of packaging. Although it has been widely studied the use of cysteine as an anti-browning agent [18,23,24,29], in the present study, it has been observed that non-treated and cysteine-treated artichokes presented similar WL; therefore, the use of cysteine on its own did not reduce the WL of fresh-cut artichoke, while its combination with EOs did. Similar results were observed in fresh-cut potatoes treated with L-cysteine, where no effect on WL was observed [55]. However, previous studies showed that the application of EOs resulted in a reduction of WL in litchi fruit [56], apricots [57], nectarines [58], grapes [59] and pears [60]. This could be related to the barrier effect that essential oils have on the fruit, reducing moisture loss and delaying dehydration of treated fruits [56].

‘Blanca de Tudela’ artichoke has been classified as a commodity with an extremely high respiration rate (RR), which is responsible for its quick deterioration and short shelf-life [61]. The RR of fresh-cut artichokes was significantly (*p*-value < 0.05) reduced in those artichokes that were treated with a combination of cysteine and EOs (Figure 3b). Fresh-cut artichokes treated with Cys + EOs 75, Cys + EOs 150 and Cys + EOs 300 after 9 days of storage presented significantly lower values of RR (227.75 ± 14.52, 232.45 ± 11.98, 211.27 ± 7.68 mg of CO_2_ kg^−1^ h^−1^) compared to the non-treated ones (311.83 ± 13.20 mg of CO_2_ kg^−1^ h^−1^). Although a previous study reported an increase in the CO_2_ concentration of artichokes treated with cysteine and EOs sealed in a PET bag [51], it needs to be considered that the present study was carried out without packaging. Thus, the RR that has been studied was the one from artichokes after being placed on a 1 L glass jar hermetically sealed for 30 min at room temperature. A reduction in the RR was observed in those artichokes treated with cysteine and EOs, an effect that could be attributed to a lower metabolism activity during its storage that has been induced by these EOs, as it has already been reported for apples [42] and table grapes [39,62]. Overall, the application of a combination of cysteine and EOs to fresh-cut artichokes resulted in an improvement in the quality parameters studied.

### 3.3. Functional Parameters of Fresh-Cut Artichokes

Artichokes are a great source of bioactive compounds for human consumption. The nutritional and pharmaceutical properties of artichokes are linked to their chemical composition, which includes high levels of polyphenolic components and inulin. Derivatives of caffeic acid are the main polyphenolic compounds in artichoke heads, with chlorogenic acid (5-O-caffeoylquinic acid) as the main one, as well as dicaffeoylquinic acid derivatives and neo-chlorogenic acid [2,61,63]. An increased tendency was observed in the total phenolic content (TPC) (Figure 4) during storage, probably due to the concentration caused by the weight loss. Significant differences (*p*-value < 0.05) between treatments were only detected in those artichokes that were treated with Cys + 150 EOs. After 4 days of storage, a significantly higher TPC (677.46 ± 37 mg of gallic acid 100 g^−1^ FW) was observed in Cys+ 150 EOs treated fresh-cut artichokes compared to the non-treated (499.49 ± 19.25 mg of gallic acid 100 g^−1^ FW). Results were in accordance with the already mentioned study carried out by Rizzo et al. [51], which observed higher levels of TPC in the presence of the applied EOs. The fact that TPC was significantly higher in those artichokes treated with a specific concentration of EOs could be due to an increase in the phenylpropanoid pathway, as previously suggested by Gutiérrez-Pozo et al. [64]. The application of Cys + 150 EOs provided the highest value of TPC compared to the rest of the treatments. As has been previously demonstrated, the application of EOs could be dose-dependent, where an excess of EOs could be harmful to the treated product [65].

The antioxidant activity of artichokes varies among cultivars, age, plant generation, growing conditions, harvest, post-harvest handling, and storage conditions [2,63,66]. The hydrophilic antioxidant activity (H-TAA) is directly linked to the TPC, as previously reported [50,66]. A similar tendency to TPC was observed when the H-TAA was studied in fresh-cut artichokes (Figure 5). After 2 days of storage, the H-TAA of fresh-cut artichokes treated with Cys + 150 EOs was the only treatment where a significantly higher value (657.9 3± 32.07) was observed compared to the non-treated (558.17 ± 36.93). However, Cys + 75 EOs and Cys + 150 EOs H-TAA were significantly higher (*p*-value < 0.05) than the rest of the treatments after 4 and 9 days of storage (Figure 5a). A similar tendency was observed in the lipophilic antioxidant activity (L-AAT) (Figure 5b), where significantly higher values were observed in those fresh-cut artichokes treated with cysteine and EOs, being in this fraction both highest concentrations of EOs, the one providing the highest values of L-TAA. The H-TAA was 15–20-fold higher than L-TAA due to artichokes, the main responsible compounds for its antioxidant potential, which are of a hydrophilic nature. In addition, hydrophilic non-phenolic compounds, such as ascorbic acid, have been found at relatively high concentrations in the edible portion of the artichoke, contributing to the H-TAA [61]. Furthermore, L-TAA could be attributed to the presence of tocopherol (vitamin E) and lutein, which have been detected in artichokes [67]. Although previous studies observed that the application of cysteine to fresh-cut artichokes did not influence the antioxidant activity [24], the results of the present study are in accordance with Rizzo et al. [44,51], where the antioxidant capacity was studied in artichoke heads treated with *Foeniculum vulgaris* EOs on its own and in combination with cysteine, achieving the highest antioxidant activity after 11 days of storage. This increase in the antioxidant activity in the presence of EOs could be explained due to the high antioxidant capacity that these EOs present, as was previously reported [32,39].

The increase in these functional phytochemicals as a result of the application of cysteine in combination with different concentrations of EOs also presents a health benefit that has been previously attributed to the bioactive content of artichokes [68,69]. Therefore, the potential use of an edible coating with cysteine and these EOs could result in fresh-cut artichokes with better quality after cold storage (2 °C) and present a higher benefit to human health.

### 3.4. Sensory Evaluation

The sensory evaluation is essential to ensure the consumer acceptability of those artichokes that were treated with cysteine and different concentrations of EOs. It could also provide key information about possible residual odour or flavour affecting fresh-cut artichokes after the post-harvest treatment with EOs. The evaluation of sensory attributes (colour, browning, dehydrated appearance, odour, sweet, bitter, firmness and general acceptability) in fresh-cut artichokes after 2, 4 and 9 days of storage was presented in Figure 6. The sensory quality, in general, presented a decreasing tendency during storage in all treatments, as has been previously reported [16,19,25]. After 2 days of storage (Figure 6a), colour and browning attributes were assessed with 3 to 5 points, where those artichokes treated with cysteine and the highest concentrations of EOs (150, 300) achieved the highest score. A similar tendency was observed after 4 and 9 days of storage at 2 °C (Figure 6b,c), observing a differentiation between both treatments with Cys + 150 EOs and Cys + 300 EOs, where colour and browning were assessed with the highest score for the intermediate concentration (Cys + 150 EOs). Similar results were observed by Rizzo et al. [51] when they treated fresh-cut artichokes with cysteine and *F. vulgare* EO.

According to the panellists, no significant differences were observed in the dehydrated appearance, firmness, and odour of fresh-cut artichokes after 2 days of storage. However, after 4 days of storage, firmness and dehydration attributes significantly decreased to a score of 3 in non-treated and cysteine-treated artichokes. At the end of the storage, Cys + 150 EOs resulted in the one presenting the highest scores of firmness and dehydration. Similar scores were attributed to odour for all the treatments, including the ones presenting the EOs. Besides, residual odour related to EOs was only detected in fresh-cut artichokes treated with the highest concentrations of EOs (150 and 300 µL) after 2 days of storage, while after 4 and 9 days of storage, no residual odour was detected due to the use of a very low concentration, a short time of treatment and their storage in open trays. Opposite results to Rizzo et al. [44], as they observed a higher intensity of fennel odour and flavour in those artichokes that were packed with a small concentration of *F. vulgare*. This difference could be attributed to the storage in hermetically sealed trays, compared to the storage in open trays in the present study. Attributes such as sweetness and bitterness of fresh-cut artichokes were also evaluated, observing similar scores between treatments, with Cys + 150 EOs the one presenting the lowest score. At last, for general acceptability, a similar tendency was observed at all the different stages of the storage period. Those fresh-cut artichokes that were treated with cysteine and 150 µL of EOs presented the highest score, followed by the highest concentration of EOs (300 µL of EOs).

## 4. Conclusions

The application of EOs in fresh-cut artichokes in combination with cysteine could significantly reduce browning, improve the overall quality, as well as increase the shelf-life of this ready-to-eat product. In the present study, it has been revealed that the use of cysteine in combination with 150 µL of EOs (thymol, eugenol and carvacrol) considerably reduced browning while maintaining quality parameters, increased shelf-life and resulted in excellent consumer acceptability. Considering that the present study was carried out in fresh-cut artichokes that were stored in open trays at 2 °C for 9 days, the use of cysteine and these EOs, in combination with an optimal packing system, could improve their quality and shelf-life.

## Figures and Tables

**Figure 1 foods-12-04414-f001:**
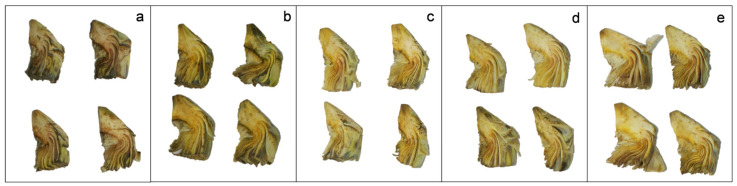
Browning of non-treated artichokes (**a**), and treated with Cysteine (**b**), Cysteine and EOs 75 µL (**c**), Cysteine and EOs 150 µL (**d**) and Cysteine and EOS 300 µL (**e**) after 9 days of storage at 2 °C.

**Figure 2 foods-12-04414-f002:**
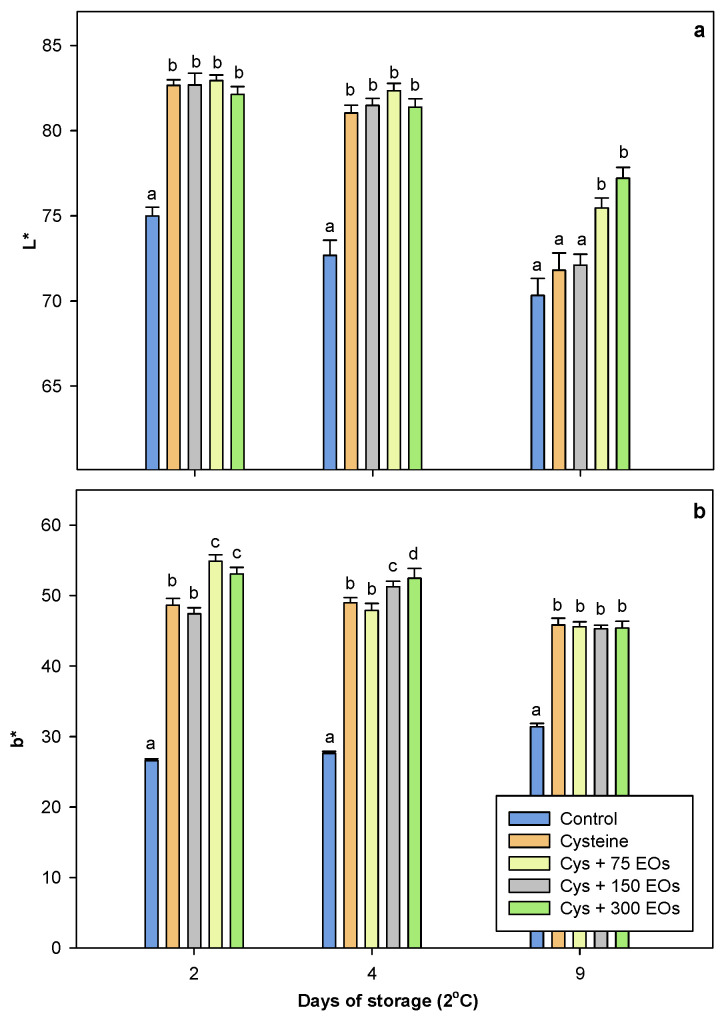
Colour parameter L* (**a**) and b* (**b**) of fresh-cut artichokes non-treated and treated with cysteine (Cys), cysteine and 75 µL of EOs (Cys + 75 EOs), cysteine and 150 µL of EOs (Cys + 150 EOs) and cysteine and 300 µL of EOs (Cys + 300 EOs) after 2, 4 and 9 days of storage at 2 °C. Significant differences (*p*-value < 0.05) between treatments are presented with different lower-case letters (Tukey’s test).

**Figure 3 foods-12-04414-f003:**
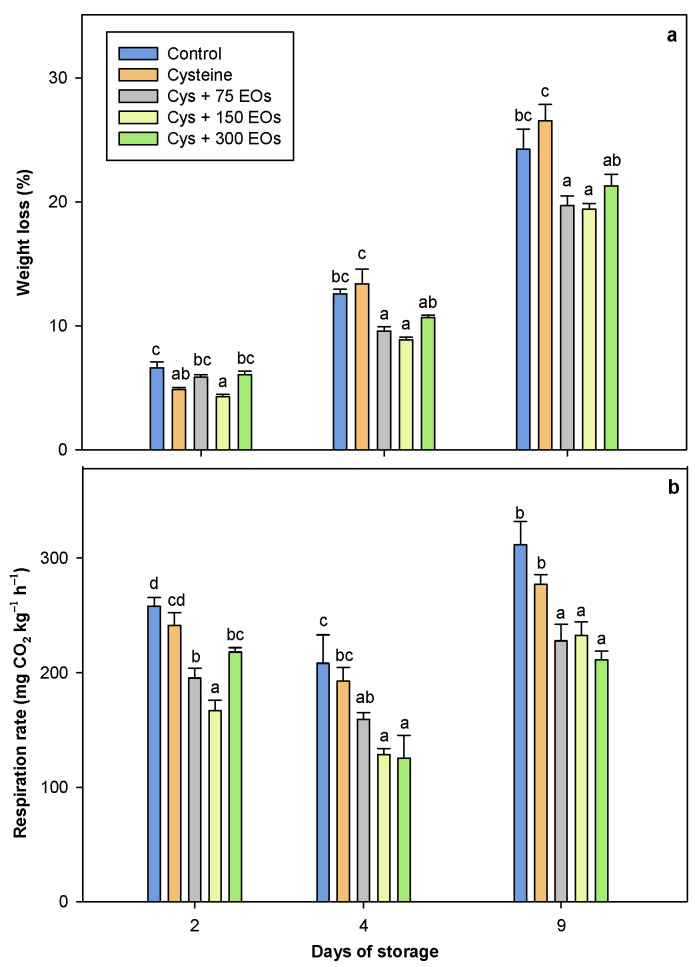
Weight loss (**a**) and respiration rate (**b**) of fresh-cut artichokes non-treated and treated with cysteine (Cys), cysteine and 75 µL of EOs (Cys + 75 EOs), cysteine and 150 µL of EOs (Cys + 150 EOs) and cysteine and 300 µL of EOs (Cys + 300 EOs) after 2, 4 and 9 days of storage at 2 °C. Significant differences (*p*-value < 0.05) between treatments are presented with different lower-case letters (Tukey’s test).

**Figure 4 foods-12-04414-f004:**
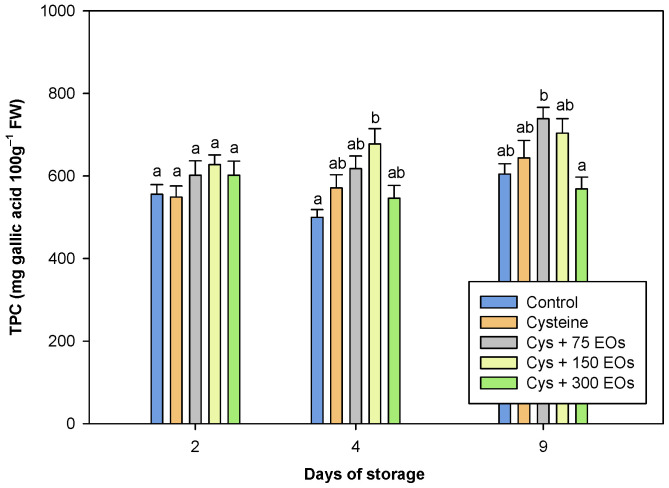
Total phenolic content (mg gallic acid 100 g^−1^) of non-treated and treated with cysteine, cysteine and 75 µL of EOs (Cys + 75 EOs), cysteine and 150 µL (Cys + 150 EOs), and cysteine and 300 µL (Cys + 300 EOs) fresh-cut artichokes after 2, 4 and 9 days of storage at 2 °C. Significant differences (*p*-value < 0.05) are presented with different lower-case letters (Tukey’s test).

**Figure 5 foods-12-04414-f005:**
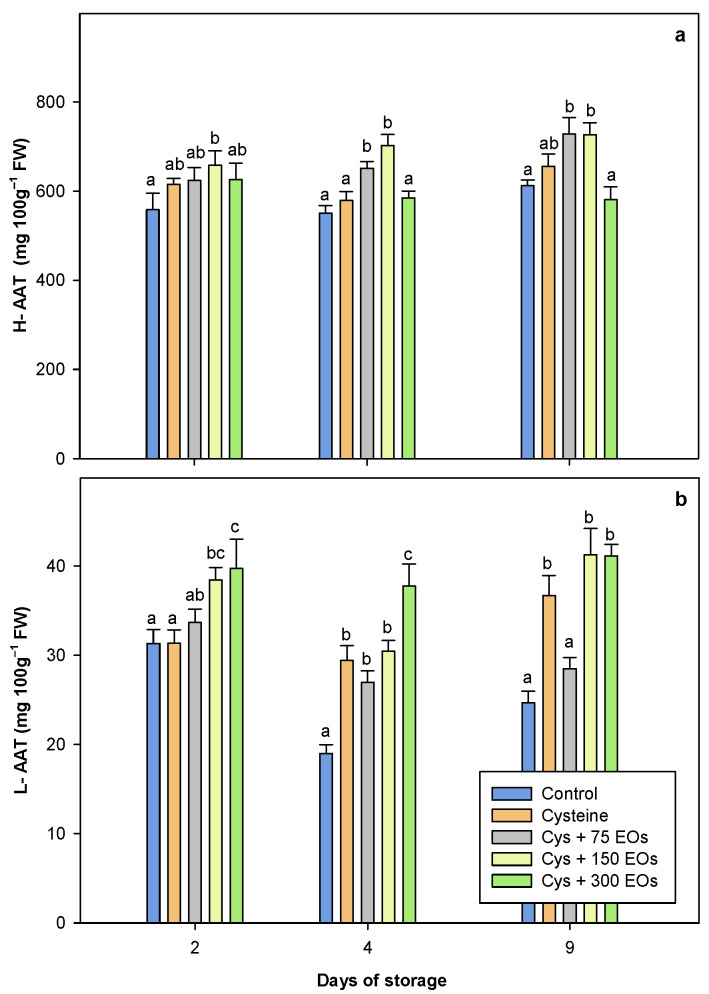
Total antioxidant activity (TAA) in the hydrophilic fraction (**a**) and lipophilic fraction (**b**) of non-treated and treated with cysteine, cysteine and 75 µL of EOs (Cys + 75), cysteine and 150 µL (Cys + 150), and cysteine and 300 µL (Cys + 300) fresh-cut artichokes after 2, 4 and 9 days of storage at 2 °C. Significant differences (*p*-value < 0.05) are presented with different lower-case letters (Tukey’s test).

**Figure 6 foods-12-04414-f006:**
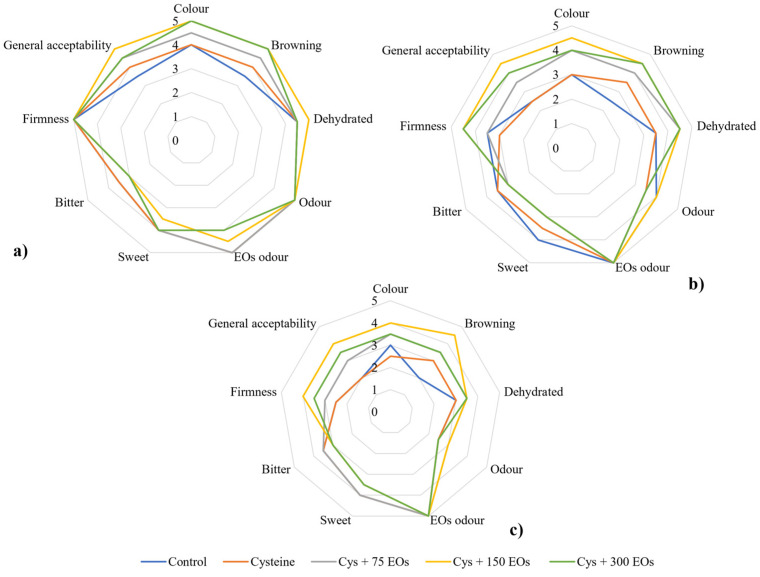
Descriptive sensory analysis of non-treated fresh-cut artichoke heads and treated with cysteine, cysteine and 75 µL of EOs (Cys + 75), cysteine and 150 µL (Cys + 150), and cysteine and 300 µL (Cys + 300) after 2 (**a**), 4 (**b**) and 9 (**c**) days of storage at 2 °C. Colour, browning, dehydrated appearance, odour, sweetness, bitterness, firmness and general acceptability were evaluated.

**Table 1 foods-12-04414-t001:** Organoleptic characteristics evaluated for fresh-cut artichokes.

Sensory Attribute	Description
Visual appearance	
Colour	Colours ranging from White to yellow
Browning in cut areas	Presence of black areas in cut zones
Dehydrated	High moisture loss
Odour	
General odour	Smell due to volatile compounds identified with a cardoon odour
EOs odour	Smell due to residual volatile compounds from the EOs applied
Taste	
Sweet	Primary tastes produced by substances
Bitter	such as sucrose (sweet), caffeine
Sour, acid	(bitter), and citric acid (sour or acid)
Texture	
Firmness	Force necessary to break the product and sound generated from frostbite
General acceptability	General impression of the sample

## Data Availability

Data is contained within the article.

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
