# Peer review of "Effect of Cysteine with Essential Oils on Quality Attributes and Functional Properties of ‘Blanca de Tudela’ Fresh-Cut Artichoke"

_foods, 2023, doi:10.3390/foods12244414_

Round 1

Reviewer 1 Report

Comments and Suggestions for Authors

The  aim of this paper, according to authors, was to evaluate the effect of L-cysteine and in combination with a mixture of essential oils (Eugenol, Thymol and Carvacrol) on browning, quality and bioactive properties of fresh-cut artichokes stored for 9 days at 2ºC. The objective is interesting, the structure is well laid out and the Introduction provides the majority of necessary information. Before proceeding to analyze my opinion, I have to report that in the manuscript uploaded Figures 3,4 and 5 are missing, and thus I cannot evaluate the quality of the submission before these figures are incorporated within the text. Figure 6 is also misplaced.  Authors need to carefully re-visit the whole text, as there are numerous points that need to be checked and corrected, in terms of language editing and syntax.

Some other points to be addressed are related to the experimental design. Authors chose a low temperature of 2ºC to perform their storage study. Has the issue of chilling injury been considered? I believe that authors have to justify the selection of both temperature and storage period (9 days), and also to include an extensive analysis of the chilling injury phenomenon in their Introduction part, as being one crucial cause of vegetable spoilage. Another objection is related to the number of measurements acquired. Why only assessing all attributes at 3 distinct time points (at 2, 4 and 9th day)? I believe that more measurements are needed, especially between the 4th and 9th day.

Other minor comments:

L11: a minimum or a maximum?

L15: why at 2ºC? See also general comments

L41-43: explain how enzymatic browning is related to total phenolic content.

L84-85: explain explicitly the purpose of the specific treatment (dipping in ice water with sodium hypoclorite).

L147: rejection by consumers

L159: significant differences

L163-165: Rephrase, check the syntax.

L184: lightness or yellowness?

L206: decreased or increased?

§3.2: How can one explain L-cystein/EO effect on weight loss? How the incorporation of those anti browning agents can affect CO2 concentration or transpiration rate? An explicit explanation is required.

Comments on the Quality of English Language

Authors need to carefully re-visit the whole text, as there are numerous points that need to be checked and corrected, in terms of language editing and syntax.

Author Response

Dear reviewer,

Thank you very much for all valuable comments. All changes have been included and highlighted in orange ink in the manuscript. 

Comment 1. The aim of this paper, according to authors, was to evaluate the effect of L-cysteine and in combination with a mixture of essential oils (Eugenol, Thymol and Carvacrol) on browning, quality and bioactive properties of fresh-cut artichokes stored for 9 days at 2ºC. The objective is interesting, the structure is well laid out and the Introduction provides the majority of necessary information. Before proceeding to analyze my opinion, I have to report that in the manuscript uploaded Figures 3,4 and 5 are missing, and thus I cannot evaluate the quality of the submission before these figures are incorporated within the text. Figure 6 is also misplaced.  Authors need to carefully re-visit the whole text, as there are numerous points that need to be checked and corrected, in terms of language editing and syntax.

The figures have been added, please accept our apologies for that, something has happened once the manuscript was submitted that these figures disappeared.

The Quality of English Language has been reviewed and highlighted in red ink.

Comment 2. Some other points to be addressed are related to the experimental design. Authors chose a low temperature of 2ºC to perform their storage study. Has the issue of chilling injury been considered? I believe that authors have to justify the selection of both temperature and storage period (9 days), and also to include an extensive analysis of the chilling injury phenomenon in their Introduction part, as being one crucial cause of vegetable spoilage.

The recommended temperatures for the conservation of fresh-cut artichokes are low temperatures. However, artichokes are not susceptible to chilling injury at 2oC, they need to be stored at temperatures lower than 0oC. Most of the studies on fresh-cut artichokes have been carried out at 2 to 4oC (El-Mogy et al. 2020).

Comment 3. Another objection is related to the number of measurements acquired. Why only assessing all attributes at 3 distinct time points (at 2, 4 and 9th day)? I believe that more measurements are needed, especially between the 4th and 9th day.

An explanation of the selection of these three time points have been included in the manuscript:

These days represented an early, mid- and late stage of the storage of fresh-cut artichokes (lines 105-106).

Comment 4. Other minor comments:

L11: a minimum or a maximum?

‘Minimum’ has been changed to ‘maximum’ and highlighted in orange ink (Line 10).

L15: why at 2ºC? See also general comments

The selection of 2oC is based on the commercialization temperature of fresh-cut artichokes. Besides, all previous studies on fresh-cut artichokes have been carried out at temperature ranges from 2 to 4oC (Rizzo et al. 2021, Rizzi et al. 2013, Gomez di Marco et al. 2011, El-Mogy 2020).

L41-43: explain how enzymatic browning is related to total phenolic content.

The full explanation has been included in the manuscript:

Enzymatic browning is considered as the main cause of quality loss in fresh-cut artichokes [3,4]. It takes place due to the high polyphenol content of artichokes, polyphenols that serves as substrate for the polyphenol oxidase (PPO). Once artichokes are subjected to wounding or cutting, a cellular disruption take place, allowing PPOs to get in contact with their substrates. These polyphenols are oxidated to quinones, resulting in dark pigment causing tissue browning on fresh-cut artichokes [5–7]. (Lines 36-42)

Three news references have also been added:

Taranto, F.; Pasqualone, A.; Mangini, G.; Tripodi, P.; Miazzi, M.M.; Pavan, S.; Montemurro, C. Polyphenol Oxidases in Crops: Biochemical, Physiological and Genetic Aspects. Int. J. Mol. Sci. 2017, 18, 377. doi: 10.3390/ijms18020377

Constabel, C.P.; Barbehenn, R. Defensive Roles of Polyphenol Oxidase in Plants. In: Schaller, A. (eds) Induced Plant Resistance to Herbivory 2008. Springer, Dordrecht. doi:10.1007/978-1-4020-8182-8_12

Ricceri, J.; Barbagallo, R.N. Role of protease and oxidase activities involved in some technological aspects of the globe artichoke processing and storage. LWT 2016, 71, 196-201, doi: 10.1016/j.lwt.2016.03.039.

L84-85: explain explicitly the purpose of the specific treatment (dipping in ice water with sodium hypoclorite).

The use of this specific treatment, prior to their dipping in the tested treatments, was to eliminate soil and other foreign residues. It has been specified in the manuscript:

After that, each artichoke heart was cut in 8 parts (1/8), mixed and dipped in ice water containing 100 ppm of sodium hypochlorite as a disinfectant solution. (Line 90)

L147: rejection by consumers

This change has been included in the manuscript in orange ink (Line 158).

L159: significant differences

This change has been included in the manuscript in orange ink (Line 169).

L163-165: Rephrase, check the syntax.

The phrase has been reviewed and syntax checked:

It has been previously reported that artichokes quality characteristics depend on cultivar, genotype, harvest and storage time (Lines 174-175)

L184: lightness or yellowness?

The parameter b* from the CIELAB colour space refers to colour from yellow to blue, as higher b* value more yellowness, while as lower b* values more greenness. Therefore, that´s the reason why we mention yellowness when we are talking about the b* parameter. While, for the L* parameter we talk about lightness, as higher values of L*, higher lightness the sample presented.

L206: decreased or increased?

‘increased’ has been included in the manuscript in orange ink (Line 214).

Section 3.2: How can one explain L-cystein/EO effect on weight loss? How the incorporation of those anti browning agents can affect CO2 concentration or transpiration rate? An explicit explanation is required.

A paragraph about this potential explanation of the effect of the EOs have been included in the manuscript:

Similar results were observed in fresh-cut potatoes treated with L-cysteine, where no effect on WL was observed [53]. However, previous studies showed that the applica-tion of EOs resulted in a reduction of WL in litchi fruit [54], apricots [55], nectarines [56], grapes [57] and pears [58]. This could be related to the barrier effect that essential oils have on the fruit, reducing moisture loss and delaying dehydration of treated fruits [56]. (lines 228-233)

The corresponding references have been added:

Li, G.; Wang, X.; Zhu, H.; Li, G.; Du, J.; Song, X.; Erihemu, X. Use of Different Food Additives to Control Browning in Fresh-Cut Potatoes. Food Sci Nutr 2023, doi:10.1002/fsn3.3714.

Ali, S.; Khan, A.S.; Malik, A.U. Postharvest L-Cysteine Application Delayed Pericarp Browning, Suppressed Lipid Peroxidation and Maintained Antioxidative Activities of Litchi Fruit. Postharvest Biol Technol 2016, 121, 135–142, doi:10.1016/j.postharvbio.2016.07.015.

Shemy, M. Effect of Some Essential Oils, Salts And Salicylic Acid On Reducing Decay, Keeping Quality And Prolonging Shelf-Life Of Canino Apricot Fruits. J. Plant Prod. 2020, 5, 125-142.

Abd, S.M.; Wahab, E. Maintain Postharvest Quality of Nectarine Fruits by Using Some Essential Oils. Middle East J. App. Sci. Tech. 2015, 5, 855-568. 

Salimi, L.; Arshad, M.; Rahimi, A.R.; Rokhzadi, A.; Amini, S.; Azzi, M. Effect of Some Essential Oils on Post Harvest Quality of Grapevine (Vitis Vinifera Cv Rasha (Siah-e-Sardasht)) during Cold Storage. Int. J. Biosci. 2013, 75–83, doi:10.12692/ijb/3.4.75-83.

Saleh, M.A.; Zaied, N.S.; Maksoud, M.A.; Hafez, O.M. Application of Arabic Gum and Essential Oils as the Postharvest Treatments of Le Conte Pear Fruits during Cold Storage. Asian J. Agric. Hort. Res. 2019, 1–11, doi:10.9734/ajahr/2019/v3i329999.

Reviewer 2 Report

Comments and Suggestions for Authors

In this paper, the authors investigated the effect of a mixture of cysteine and essential oil compounds (eugenol, thymol and carvacrol) on the browning, quality and bioactive compounds of freshly cut artichokes stored at 2 ºC for 9 days. The manuscript is not of sufficient quality to be published in the journal FOODS.

As I understand it, the authors used essential oil compounds and not essential oils, although it is written throughout the manuscript that essential oils were used. The authors did not explain why they chose these particular compounds or their ratio in the mixture. Also, no individual components with cysteine were tested, only a combination of compounds with cysteine, so it remained unclear which compounds contributed to the activities reported by the authors and to what extent.

In the Material and methods, the concentration of cysteine in the control group or in the groups with other components is not stated, nor is the concentration of thymol

Specify cold storage temperature

Line 118 Why was it done in duplicate?

Lines 154-156 are not described in the Material and Methods. How did you measure these two parameters and in what units?

Delete lines 173-176

Figures 3, 4 and 5 are missing

Author Response

Dear reviewer,

Thank you very much for all valuable comments. All changes have been included and highlighted in green ink in the manuscript. 

Comment 1. Some citations (references) in the material and methods section need strict review so that the authors focus on them appropriately. As I understand it, the authors used essential oil compounds and not essential oils, although it is written throughout the manuscript that essential oils were used.

The citations of the material and methods section has been reviewed and updated, changes have been highlighted in green ink.

It has been specified the use of essential oil compounds in the introduction:

Previous studies have elucidated the potential antioxidant effect that essential oils major components (EOs) have when they are applied to fruits and vegetables, at different stages of the food chain [30,31].  (lines 61-63)

Besides in the M&M section a phrase specifying the use of Essential oils compounds was included:

The essential oils compounds (EOs) for this study were eugenol, thymol and carvacrol. (lines 90-91)

Comment 2. The authors did not explain why they chose these particular compounds or their ratio in the mixture. Also, no individual components with cysteine were tested, only a combination of compounds with cysteine, so it remained unclear which compounds contributed to the activities reported by the authors and to what extent.

This explanation has been added in the manuscript in the introduction section:

It has been previously demonstrated the synergic effect that major components of essential oils (eugenol, thymol and carvacrol) have on the fungal spoilage of fruit [38,41]. (lines 67-69).

And the corresponding citations added to the references section:

Nikkhah, M.; Hashemi, M.; Najafi, M. B. H.; Farhoosh, R. Synergistic effects of some essential oils against fungal spoilage on pear fruit. Int. J. Food Microbiol. 2017, 257, 285–294. doi: 10.1016/j.ijfoodmicro.2017.06.021

Comment 3. In the Material and methods, the concentration of cysteine in the control group or in the groups with other components is not stated, nor is the concentration of thymol

The concentration of the cysteine and EOs have been included:

All hearts slices were drained and 72 different 1/8 parts were randomly selected for each different treatment and replicates and dipped in distilled water (Control), L-Cysteine at 0.028M at a pH of 2.2 (Cys), Cysteine + 75 µL/L EOs (Cys + 75 EOs: 25 µL Eugenol + 25 µL Thymol + 25 µL Carvacrol), Cysteine + 150 µL/L EOs (Cys + 150 EOs: 50 µL Eugenol + 50 µL Thymol + 50 µL Carvacrol), Cysteine + 300 µL/L EOs (Cys + 300 EOs: 100 µL Eugenol + 100 µL Thymol + 100 µL Carvacrol). The reagents used were L-cysteine hydrochloride monohydrate, carvacrol (>98%), eugenol (>98%) and thymol (98.5%) (Sigma Aldrich, Madrid, Spain) at the specific concentrations diluted in distilled water. (lines 91-99).

Comment 4. Specify cold storage temperature         

Fresh-cut artichokes were stored at 2 ⁰C and 85% of Relative Humidity, as it is stated in line 100 an 101:  ‘Lots of artichoke hearts were dried at room temperature and stored in polyethylene (PE) opened trays at 2 ⁰C and 85% RH for 9 days.’

Besides, the cold temperature has been specified every time it was mentioned in the manuscript (lines 105, 317 and 340). It has also been included in the X-axis of some of the figures, the have been highlighted in green ink. 

Comment 5. Line 118 Why was it done in duplicate?

It was carried out in duplicate to verify that the protocol was working. In previous studies, such as Gimenez-Berenguer et al. 2022 in artichoke was also carried out in duplicate.

Comment 6. Lines 154-156 are not described in the Material and Methods. How did you measure these two parameters and in what units?

The browning evaluation methodology is presented in the Material and methods section in line 114 to 117 : ‘Browning of artichokes was evaluated individually based on colour parameters (L*, a* and b*), using a Minolta colourimeter (CRC200; Minolta, Osaka, Japan) at three points of the external surface for each artichoke part using the CIE Lab System. Results were expressed as the mean ± SE.’

The parameter b* and L* are from the CIE Lab system and they don´t have units. 

Comment 7. Delete lines 173-176

These lines have been deleted.

Comment 8. Figures 3, 4 and 5 are missing

The figures have been added, please accept our apologies for that, something has happened once the manuscript was submitted that these figures disappeared.

Besides the English Quality has been reviewed and changes have been highlighted in red ink.

Reviewer 3 Report

Comments and Suggestions for Authors

Some suggestions are in the manuscript.

Some citations (references) in the material and methods section need strict review so that the authors focus on them appropriately.

Several figures were missing from the results section. Which prevents a robust and focused review.

The references section needs a lot of attention. Recent references are few (2017 to 2023).

In general, the work is interesting and could be a useful piece of knowledge, but as it stands I cannot recommend it for publication. There are some minor and major issues that should be corrected.

Comments on the Quality of English Language

In general, the quality of English is quite good, but there are fue places that need a lot of attention

Author Response

Dear reviewer,

Thank you very much for all valuable comments. All changes have been included and highlighted in blue ink in the manuscript. 

Comment 1. Some citations (references) in the material and methods section need strict review so that the authors focus on them appropriately.

The citations in the Material and methods section have been reviewed and changes highlighted in green ink.

Comment 2. Several figures were missing from the results section. Which prevents a robust and focused review.

The figures have been added, something has happened once the manuscript was submitted that these figures disappeared. Please accept our apologies for that.

Comment 3. The references section needs a lot of attention. Recent references are few (2017 to 2023).

The references section has been reviewed and changed made and included in blue ink. Few recent references have been added in lines 49 and 59. And both recent references have been added to the reference section:

El-Mogy, M.M.; Parmar, A.; Ali, M.R.; Abdel-Aziz, M.E.; Abdeldaym, E.A. Improving Postharvest Storage of Fresh Artichoke Bottoms by and Edible Coating of Cordia myxa gum. Postharvest Biol Technol 2020, 163, 111143, doi: 10.1016/j.postharvbio.2020.111143

Pompili, V., Mazzocchi, E., Moglia, A. et al. Structural and expression analysis of polyphenol oxidases potentially involved in globe artichoke (C. cardunculus var. scolymus L.) tissue browning. Sci Rep 2023, 13, 12288. https://doi.org/10.1038/s41598-023-38874-4

Comment 4. Suggestions added in the manuscript

Suggestion added in the manuscript have been added and highlighted in blue ink.

Besides the English Quality has been reviewed and changes have been highlighted in red ink.

Round 2

Reviewer 2 Report

Comments and Suggestions for Authors

  No comments

Author Response

Thank you!

Reviewer 3 Report

Comments and Suggestions for Authors

This comment need a lot of attention!!!!

Author comment 1. Some citations (references) in the material and methods section need strict review so that the authors focus on them appropriately.

The citations in the Material and methods section have been reviewed and changes highlighted in green ink.

Author Response

Thank you very much again for your kind comments. Your changes have been included and highlighted in blue ink.

The references suggested has been added in lines 126 to 136:

The reaction was carried out in duplicate following Re et al. [46] protocol with some modifications as presented in Giménez et al. [47] , and results were expressed as the mean ± SE of mg of Trolox equivalent per 100 g of fresh weight (FW).
2.4. Total phenolic content
Total Phenolic Content (TPC) was measured following Swaim and Hillis [48] protocol with slight modifications, as previously mentioned by Martínez-Esplá et al. [49]. A water:methanol (2:8) solution containing 2 mM NaF was used and extracts quantified using the Folin–Ciocalteu reagent. Results (mean ± SE) were expressed as mg of gallic acid equivalent per 100 g FW.

And included in the references section:

46. Re, R.; Pellegrini, N.; Proteggente, A.; Pannala, A.; Yang, M.; Rice-Evans, C. Antioxidant Activity Applying an Improved ABTS Radical Cation Decolorization Assay. Free Radic Biol Med 1999, 26, 1231–1237, doi:10.1016/S0891-5849(98)00315-3.
48 Swain, T.; Hillis, W.E. The Phenolic Constituents of Prunus Domestica . I.—The Quantitative Analysis of Phenolic Constituents. J Sci Food Agric 1959, 10, 63–68, doi:10.1002/jsfa.2740100110.

Besides, the change of citations is due to the addition of new references in the introduction section, therefore the numbers have been moved. If you checked the references, they are still the same, the numbers have changed but references are still the same.
